# MicroRNA-488-3p-loaded engineered exosomes inhibit proliferation, migration and invasion of hepatocellular carcinoma by targeting SEC61G

Huijie Gao[1,2], Zhaobin He[1,2], Shengbiao Yang[1,2], Xiqiang Wang[1,2], Qisen Yan[1,2], Chao Gao[1,2], Naiqing Liu[3], Zhaoyang Zhang[2,4], Weibo Niu[1,2], Jun Niu[1,2], Cheng Peng[1,2]*

1 Department of Hepatobiliary Surgery, General Surgery, Qilu Hospital, Cheeloo College of Medicine, Shandong University, Jinan, Shandong, China, 2 The Institute of Laparoscopic Minimally Invasive Surgery of Shandong University, Jinan, Shandong, China, 3 Department of General Surgery, Linyi Central Hospital, Linyi, Shandong, China 4 Department of Emergency Medicine, Qilu Hospital, Cheeloo College of Medicine, Shandong University, Jinan, Shandong, China

* Dr.Peng@email.sdu.edu.cn

## Abstract

SEC61G is an oncogene in hepatocellular carcinoma (HCC), a common malignant tumor worldwide. MicroRNAs (miRNAs) regulation of oncogenes are available therapeutic strategies being investigated in HCC, but the effective miRNA delivery remains a challenge. Here, we investigated the potential therapeutic effects of miRNA-loaded engineered exosomes in patients with HCC. MiRNAs that could bind to SEC61G were screened using Targetscan, and were verified using HepG2 cells viability after transfecting miRNAs mimic. Five miRNAs binding to SEC61G,among which, miR-651-3p and miR-488-3p mimic significantly inhibited HepG2 cells viability (p < 0.05) and decreased SEC61G protein expression. Then, dual-luciferase reporter assay also confirmed SEC61G as a target of miR-488-3p in HCC. After that, miR-488-3p-loaded engineered exosomes (Exo-miR-488-3p) were isolated from the supernatant of 488-3p-overexpressed cells and identified via nanoparticle tracking analysis, transmission electron microscopy and western blot. In vitro experiments showed that exo-miR-488-3p significantly inhibited proliferation, colony formation, migration and invasion of HepG2 cells than corresponding negative control. In addition, Exo-miR-488-3p tended to induce HepG2 cells apoptosis, though this relationship was not statistically significant. In conclusion, exo-miR-488-3p inhibits the malignant cytological activities in HCC, a possible strategy in the treatment of HCC.

## Introduction

Liver cancer is a common malignant tumor and a major cause of mortality caused by cancer in the world [1]. Global Cancer Statistics revealed that 905,677 new cases (4.7%, the sixth commonly diagnosed cancer) and 830,180 deaths (8.3%, the third

**Data availability statement:** All relevant data are within the manuscript and its Supporting Information files.

**Funding:** This work was supported by the National Natural Science Foundation of China, grant number 82072674; and the Natural Science Foundation of Shandong Province, grant number ZR2020MH258.The funders had no role in study design, data collection and analysis, decision to publish, or preparation of the manuscript.

**Competing interests:** The authors have declared that no competing interests exist.

cause of cancer deaths) were related to liver cancer in 2020 [2]. These estimates will further increase by 56.4% in 2040 [3]. Hepatocellular carcinoma (HCC) constitutes 90% of liver cancers [1]. Most cases of HCC are confirmed at later stages because of their asymptomatic nature in early stages [4]. Despite remarkable advancements in the treatment of HCC, its prognosis remains poor, with minimal benefits [5,6].

Small non-coding RNAs, microRNAs (miRNAs), are genetic modulators involved in many cellular pathways and regulate the expression of various target genes [7]. Thus, miRNAs are an interesting therapeutic option that regulate cellular processes [8]. Exosomes are a subset of extracellular vesicles present in cell-cell communications [9]. Exosomes can transfer functional molecules, such as miRNAs, mRNAs and enzymes, to distant or neighboring cells, ultimately affecting the activities of these cells [10–12]. They are considered as endogenous nanomaterials/carriers for drug delivery and therapy [13]. For example, tumor cell-derived exosomal miR-25 is involved in tumorigenesis of HCC by decreasing SIK1 expression [14]. Exosomal miR-141 promotes metastasis of ovarian cancer by activating the YAP1/GROα/CXCRs signaling cascade [15]. Exosomal miR-205 promotes tumorigenesis and tamoxifen resistance in breast cancer by targeting E2F1 o therapy [16].

Protein Transport Protein, SEC61 Gamma Subunit (SEC61G), is a component of the protein translocation apparatus of the endoplasmic reticulum (ER) membrane [17]. SEC61G dysregulation is involved in many cancers, including cervical cancer [18], breast cancer [19], kidney carcinoma [20], lung adenocarcinoma [21], glioblastoma [22], and head and neck squamous cell carcinoma [23]. However, there is paucity of data on the role of SEC61G in HCC. Our previous study revealed that expression of SEC61G is highly elevated in HCC, and it acts as an oncogene in the pathogenesis in HCC [24].

We hypothesized that targeting SEC61G exerts therapeutic effects on the progression and metastasis of HCC. To test this hypothesis, we identified miRNAs that could target SEC61G using Targetscan, and further selected and validated them through *in vitro* experiments. MiR-488-3p was finally selected. Since miRNA is easily degraded by direct infusion, miR-488-3p-loaded engineered exosomes were prepared for miR-488-3p delivery to HCC cells for subsequent experiments. We found that miR-488-3p-loaded engineered exosomes retarded the progression of HCC by targeting SEC61G.

## Materials and methods

### Cell culture and transfection

Two cell lines from the ATCC collection (Rockville, MD, USA) were used in this study, including HCC HepG2 cells and human embryonic kidney 293T cells. In an incubator under the conditions of 37°C and 5% $CO_2$, cells were cultured in Dulbecco's Modified Eagle Medium (DMEM) (#C11995500BT, Gibico, Ground Island, NY, USA), and were supplemented with 10% fetal bovine serum (FBS) (#FSP500, ExCell Bio Technonlogy Co., Ltd., Shanghai, China) and penicillin-streptomycin (penicillin 100 U/mL and streptomycin 0.1 mg/mL, Sigma, St Louis, MO, USA). The cells were transfected when they reached the logarithmic phase, and two hours prior to transfection, the

medium was changed to fresh complete culture medium. Afterwards, as per the manufacturer's protocol, miRNAs mimics and corresponding negative control (NC) transfection into HepG2 cells were conducted using Lipofectamine 2000 (Invitrogen, California, USA). Six hours after transfection, the medium was changed to fresh complete culture medium.

### Dual-luciferase reporter assay

To demonstrate the direct interaction of miRNAs with SEC61G, the wild type (WT) or mutant (MUT) 3'-UTR fragments of SEC61G containing the binding sites were amplified and cloned into the psiCHECK™-2 vectors. Then, miRNAs mimic or mimic NC and psiCHECK™-2 SEC61G WT/MUT transfection into 293T cells was conducted utilizing Lipofectamine 2000 Reagent (Invitrogen). A Dual-Luciferase Reporter Assay System (#E1910, Promega Corp., Madison, WI, USA) was applied to detect the luciferase activity 48 h after transfection, and the binding intensity between miRNAs and SEC61G was reflected by the firefly luciferase activity in normalizing against Renilla luciferase activity.

### Preparation for miR-488-3p-loaded engineered exosomes

Overexpression of miR-488-3p in 293T cells was conducted using the Lentivirus packaging system, which composed of three plasmids (Oligobio, Beijing, China); pCDH-CMV-MCS-EF1-copGFP-T2A-Puro for insertions, pMD2.G for envelop and pspAX2 for package. The plasmid for insertions was digested using XbaI and NotI (New England Biolabs, UK). Target fragments were obtained using PCR amplification with the following primers: LV-pCDH-miR-488-3p-Xba/Not-F: 5'- ACCTCCATAGAAGATTCTAGAGGCTCTATGGAACAAAACCAGTTT-3', LV-pCDH-miR-488-3p-Xba/Not-R 5'-GATCGCAGATCCTTCGCGGCCGCAGGGGTACATATAGTATCCATCTTTTCA-3'. The obtained target fragments were then ligated using a BM seamless cloning kit (#CL116–02, Beijing biomed Gene Technology Co., LTD., China). Lentivirus particles were generated by transfecting 293T cells with pCDH-CMV-MCS-EF1-copGFP-T2A-Puro (5 µg), pspAX2 (3.75 µg) and pMD2.G (1.25 µg) using LipoX Plus Reagent (#Oligo1002, Oligobio, Beijing, China). Six hours after transfection, the medium was changed to 10% fetal bovine serum-contained fresh complete culture medium. Moreover, lentivirus particles were collected and purified (viral titer $1.0 \times 10^9$ TU/mL) 48 h after transfection. The 293T cells were cultured in 10% FBS-contained DMEM at 37°C with 5% $CO_2$. The obtained lentivirus particles were transfected to 293T cells to obtain miR-488-3p-overexpressed 293 T cells, and then miR-488-3p-loaded engineered exosomes were isolated from the cell supernatant.

### Exosomes isolation and identification

Exosomes in the cell supernatant (miR-488-3p-overexpressed 293 T cells) were extracted through ultracentrifugation. In brief, cell supernatant was successively centrifuged at $2000 \times g$ for 15 min and $10,000 \times g$ for 30 min in order to remove dead cells and subcellular component. Next, the exosomes were further centrifuged at $120,000 \times g$ for 70 min to concentrate exosomes at the bottom of the tube. Exosomes was then rinsed and re-suspended with PBS solution, and then centrifuged at $120,000 \times g$ for 70 min. Next, the purified exosomes were re-suspended in particle-free PBS (filtration by 0.22µm water phase filter membrane). The particle concentration and size were determined using nanoparticle tracking analysis (NanoSight NS300, Malvern Panalytical). The exosomes were then stored at −80 °C for use.

Transmission electron microscopy (TEM, JEOL, Japan) was used to analyze the morphology of exosomes, while the Zetaview PMX 110 nanoparticles tracking analysis system (Particle Metrix, Germany) was used to characterize the size distribution. In addition, exosomes were further identified by detecting the expression of its markers CD9 (#60232–1-Ig, Proteintech Group, Inc., Chicago, USA), HSP70 (#66183–1-Ig, Proteintech) and CD81 (#66866–1-Ig, Proteintech) using Western blot.

### Exosome labeling

To analyze the entry of exosomes in HepG2 cells in vitro, 10 µg exosomes were re-suspended in the mixed solution of 200 µL Diluent C buffer and 2 µL PKH67 dye liquor (#MIDI67−1KT, Sigma-Aldrich), and incubated away from light for 5

min at room temperature. The PKH-67-labelled exosomes were collected after 70 min centrifugation at 120,000 × g, then were re-suspended in 2% complete culture solution (prepared without exosome serum). HepG2 cells were placed into 24-well plates, then PKH-67-labelled exosomes were added to each well when HepG2 cells were 60% confluent. After using paraformaldehyde to fix HepG2 cells for 15 min, the cells were stained using DAPI (#KGA215-50, KeyGEN Bio-TECH, Jiangsu, China), and were observed under a fluorescence microscope (#IX81, Olympus, Tokyo, Japan).

## Quantitative real-time PCR (qRT-PCR)

Total RNA isolation was conducted with the aid of a TRIzol Reagent (#CW0580, Cwbio, Jiangsu, China). We performed reverse transcription to synthesize cDNA through HiFiScript cDNA Synthesis Kit (#CW2569, Cwbio) as per the manufacturer's instructions. Then, the obtained cDNAs were then amplified with the aid of UltraSYBR Mixture (#CW0957, Cwbio) on a H9800 real-time PCR system (Suzhou Hehui Biotechnology Co., LTD., China). Relative expression was analyzed using $2^{-\Delta\Delta Ct}$ methods. The primers used were synthesized using GENEWIZ Biotechnology Co., LTD. (Suzhou, China), and the primers sequence were: miR-488-3p: Reverse transcription primer 5'-GTCGTATCCAGTGCAGGGTCCGAGGTATT CGCACTGGATACGACGACCAA-3', F-5'-GCGCGTTGAAAGGCTATTTC-3' and R-5'-AGTGCAGGGTCCGAGGTATT-3'. U6: Reverse transcription primer 5'-GTCGTATCCAGTGCAGGGTCCGAGGTATTCGCACTGGATACGACAAAATATG-3', F-5'-GTGCTCGCTTCGGCAGCACATAT-3' and R-5'-AGTGCAGGGTCCGAGGTATT-3'.

## Western blot

Cells lysis was conducted using RIPA lysis buffer, and were electrophoresed on 15% SDS-PAGE gels. After transferring them to PVDF membranes (Millipore, USA), the proteins were then blocked using 5% defatted milk-contained TBST solution. Afterwards, we co-incubated them with anti-SEC61G (#11147–2-AP, 1:1000, Proteintech), and anti-GAPDH (#60004–1-Ig, 1:5000, Proteintech) overnight at 4 °C. Subsequently, we rinsed them thrice using TBST, and co-incubated them with HRP-labeled secondary antibodies for 2 h at room temperature. The visualization of protein bands was performed using ECL-plus™ kit (Amersham, Piscataway, NJ, USA), which was then calculated using ImageJ software.

## Cell counting kit-8 (CCK8) assay

Viability of HepG2 cells in each group were evaluated using CCK-8 assay (#HY-K0301, MedChemExpress, New Jersey, USA) as the manual described. In short, after seeding cells into a 96-well plate under the density of 2000 cells/well, CCK-8 regent (10 μL) was added to each well at several time points (0, 24, 48, 72 and 96 h). After 4 h of incubation, absorbance of cells was recorded at 450 nm using a V-microplate reader (F50, Tecan, Mennedoff, Switzerland).

## Transwell assays

We performed cell invasion and migration assays using a 24-well Transwell chambers. Moreover, $5 \times 10^4$ cells were placed onto the matrigel-precoated or non-precoated upper chambers for invasion or migration assays, respectively. Meanwhile, 600 μL 10% FBS-contained culture medium was placed on the chamber at the bottom. Cells on the upper chamber were wiped off after 24 h incubations at room temperature. Afterwards, paraformaldehyde (Sangon, Shanghai, China) was used to fix the invaded or migrated cells for 15 min, and 0.1% crystal violet was used to stain these cells for 5 min. Eventually, the cells were viewed under a fluorescence microscope (#CKX-51, Olympus) and were counted using Image J.

## Wound-healing assay

The cells were placed in 6-well plates with a density of $5 \times 10^5$ cells/well and grew until 90% confluence in serum-free medium. Next, a sterile pipette tip was applied to scratch the cell monolayer to create 0.5–1 mm-wide wounds. After rinsing thrice with PBS to remove scratched cells, we incubated them at 37°C was conducted after adding serum-free

medium. At different time points, the movement of cells was viewed below an Olympus microscope, and the wounds calculated by means of Image J.

## Cell apoptosis

We detected cell apoptosis using flow cytometry. In brief, cells were first suspended with binding buffer (500 μL), and we added Annexin V/FITC (5 μL) to form a mixture. Afterwards, we added propidium iodide (5 μL). After incubating away from light for 5–15 min at room temperature, we detected apoptotic cells using a flow cytometer (FACSCalibur, BD, USA).

## Colony formation assays

Cells under a density of 1000 cells per well were placed into 6-well plates and were cultured for 14 days. Then, the medium was changed every three days and the cell state was observed. After rinsing with PBS, cell colonies were fixed for 30 min by adding 5 mL 4% paraformaldehyde (Sangon), and then stained for 5 min by adding 500 μL 0.1% crystal violet (Sangon). Finally, cell colonies were visualized under a fluorescence microscope (#CKX-51, Olympus) and were counted.

## Statistical analysis

Data were presented as mean ± standard deviation and were analyzed using SPSS (version 18.0, SPSS, Inc.). Student's t test was used to compare variables between groups. $P < 0.05$ were considered statistically significant.

## Results

### Screening and validation of miRNAs targeting SEC61G in HCC

The miRNAs that could bind to SEC61G were screened using Targetscan, thereby screening five miRNAs; miR-216b-3p, miR-342-3p, miR-651-3p, miR-488-3p and miR-570-3p. In our previous study, SEC61G was highly expressed in HCC cells, and its knockdown inhibited cell viability and triggered apoptosis of tumor cells [24]. Therefore, mimics of these miRNAs and corresponding NC were transfected into HepG2 cells to detect their effects to cell viability. In comparison with NC, cell viability of HepG2 cells decreased after transfection with mimics of miR-216b-3p, miR-651-3p and miR-488-3p. Amongst these, cell viability of HepG2 cells were significantly reduced at 72 h and 96 h after transfection with mimics of miR-651-3p and miR-488-3p (Fig 1A). Moreover, expression of SEC61G in HepG2 cells after transfection with miRNAs mimics through Western blot analysis revealed that the protein level of SEC61G in HepG2 cells reduced following transfection with mimics of miR-651-3p and miR-488-3p (Fig 1B), indicating that miR-651-3p and miR-488-3p binds to SEC61G in the pathogenesis of HCC.

### SEC61G is the target gene of miR-488-3p

We performed luciferase assay to further verify the binding of miR-651-3p and miR-488-3p with SEC61G. The intensity of Luciferase activity of miR-651-3p mimic did not significantly differ with that of mimic-NC in both SEC61G-WT and SEC61G-MUT groups (Fig 1C). Notably, in comparison with mimic-NC, miR-488-3p mimic markedly decreased (decreased to 54.54% of mimic NC, $p < 0.05$) the intensity of luciferase activity in the SEC61G-WT group, though no significant difference in the intensity in the SEC61G-MUT group (Fig 1D). This shows that SEC61G is a target of miR-488-3p.

### Isolation and identification of miR-488-3p-loaded engineered exosomes

Our previous study demonstrated that the inhibition of SEC61G repressed cell proliferation and induced apoptosis of tumor cells in HCC [24]. In conducted to verify whether SEC61G could be targeted by exosomes-transmitted miR-488-3p, miR-488-3p-loaded engineered exosomes (Exo-miR-488-3p) were prepared. Briefly, miR-488-3p-overexpressed 293 T

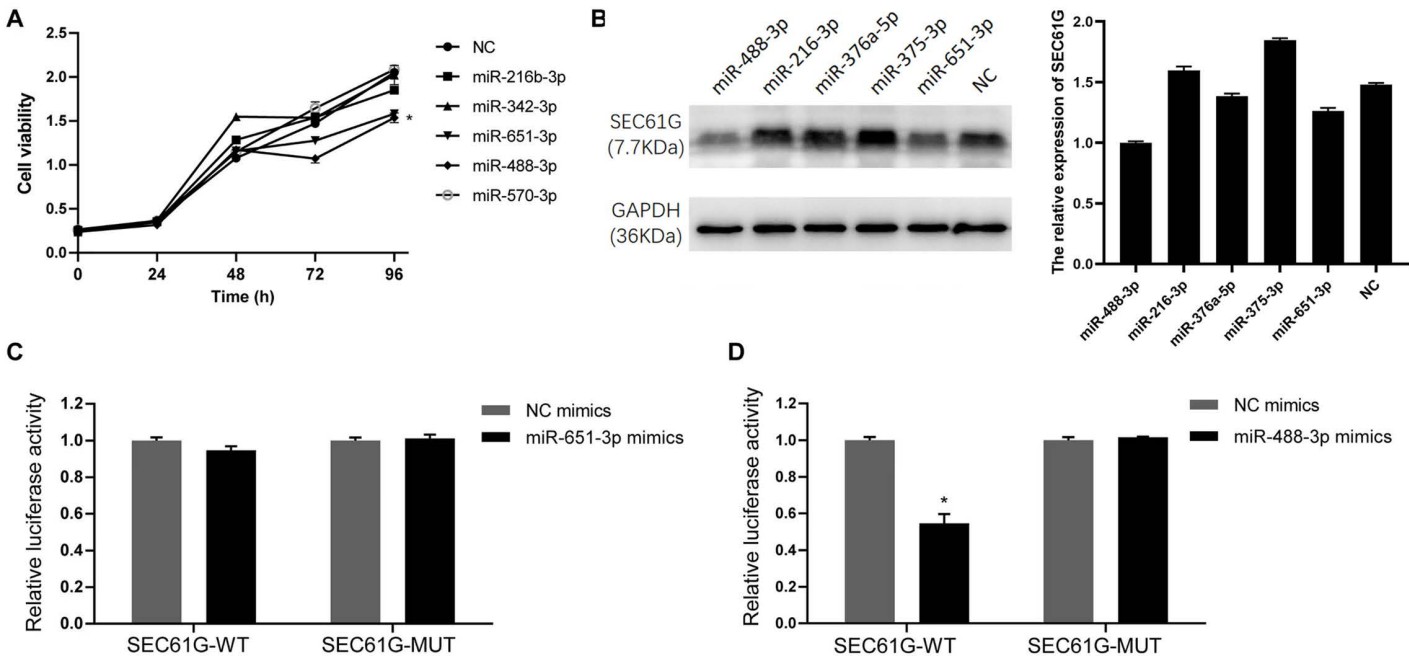

**Fig 1. Screening and validation of miRNAs binding to SEC61G.** A: Cell viability of HepG2 cells was assessed by CCK-8 assay after transfection with the indicated miRNA mimics. Data are presented as the mean ± SD from three independent experiments. B: Representative western blot protein bands (left) and quantitative expression (right) of SEC61G in HepG2 cells after transfection with miRNAs mimics. Data are presented as the mean ± SD from three independent experiments. C: Dual-luciferase reporter assay of interactions between miR-651-3p and SEC61G; D: Dual-luciferase reporter assay for determining the interactions between miR-488-3p and SEC61G. *P < 0.05.

cells were firstly prepared by transfecting miR-488-3p lentivirus particles into 293 T cells, and Exo-miR-488-3p was then isolated from cell supernatant using ultracentrifugation. TEM was applied to depict the morphological features of the isolated exosomes, and their round or elliptical shape with complete membrane structure were observed (Fig 2A). Nanoparticles tracking analysis for evaluating the size distribution of exosomes indicated a diameter ranging from 30–200 nm, with a predominant size of 130 nm (Fig 2B). Expression of exosomal markers was also monitored using western blot for further exosomes identification, and the isolated exosomes showed positive expression of HSP70, CD9 and CD81 (Fig 2C). These findings suggested that exosomes were successfully isolated. To further confirm overexpression of miR-488-3p in isolated exosomes, we performed PCR. As displayed in Fig 2D, expression of miR-488-3p was significantly elevated in Exo-miR-488-3p compared to that of Exo-NC. Thus, Exo-miR-488-3p was successfully prepared.

## Exo-miR-488-3p inhibited HepG2 cells proliferation

To demonstrate the susceptibility of HepG2 cells to miR-488-3p-overexpressed exosomes, exosomes were fluorescently labeled and co-incubated with HepG2 cells. As expected, miR-488-3p-overexpressed exosomes entered HepG2 cells after 8 h co-incubation, and the number of exosomes entering into HepG2 cells increased with time (Fig 3). CCK-8 assay indicated that viability of HepG2 cells decreased at 72 h and 96 h after co-incubation with Exo-miR-488-3p compared with co-incubation with Exo-NC (Fig 4A). Moreover, compared to Exo-NC, co-incubation with Exo-miR-488-3p markedly declined the colony formation ability of HepG2 cells (Fig 4B-C). These findings revealed that Exo-miR-488-3p hampered proliferation of HepG2 cells. Moreover, the effect of Exo-miR-488-3p on apoptosis of HepG2 cells was also detected. The number of apoptotic HepG2 cells was significantly elevated in Exo-miR-488-3p group in comparison with that of Exo-NC and control groups (Fig 4D-E).

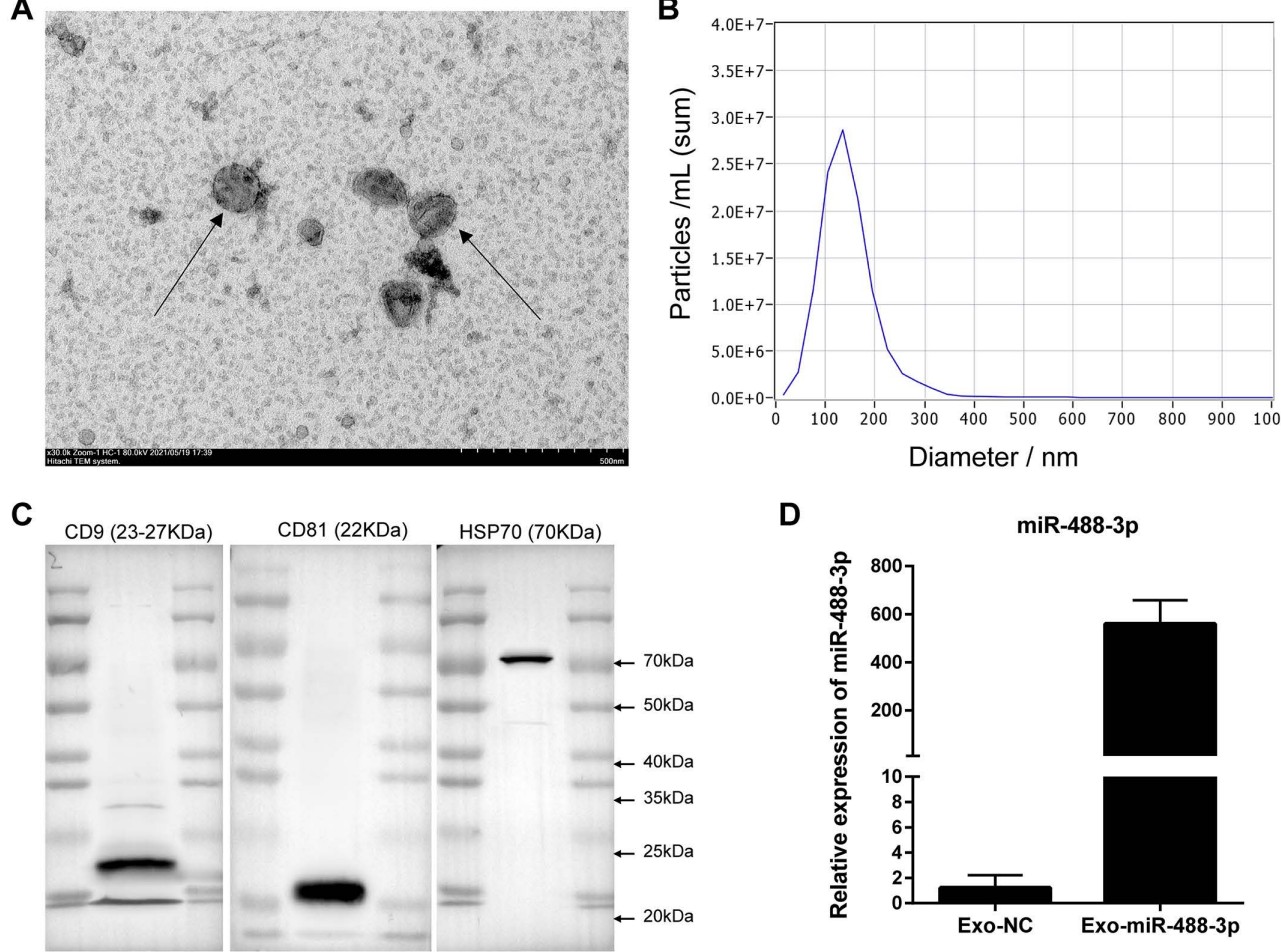

**Fig 2. Features of isolated exosomes.** A: Representative images of exosomes morphology photographed under transmission electron microscope, scale bar = 500 nm; B: The size distribution of exosomes from nanoparticles tracking analysis; C: Expression of exosome markers, CD9, CD81 and HSP70, determined by western blot; D: Expression of miR-488-3p in Exo-NC and Exo-miR-488-3p.

### Exo-miR-488-3p inhibited migration and invasion of HepG2 cells

The effect of Exo-miR-488-3p on migration and invasion of HepG2 cells was evaluated using Transwell and wound-healing assays. Compared to the Exo-NC and control groups, HepG2 cells co-incubated with Exo-miR-488-3p exhibited repressed migration ability that significantly reduced the number of migrated cells (Fig 5A). Similarly, the invasion ability of HepG2 cells were also hampered after co-incubation with Exo-miR-488-3p, and the number of cells that invade across the matrigel was markedly decreased in Exo-miR-488-3p in comparison with Exo-NC and control groups (Fig 5B). Moreover, HepG2 cells co-incubated with Exo-miR-488-3p exhibited hampered wound healing ability than Exo-NC and control groups (Fig 5C), thereby confirming the inhibitory effect of Exo-miR-488-3p on HepG2 cells migration.

### Discussion

HCC is a common malignancy, ranking as the third cause of deaths due to cancers, and the morbidity and mortality are projected to increase between 2020 and 2040 [3]. Despite the considerable progress in diagnosis and treatment, the overall survival of HCC patients is still dissatisfactory, revealing the need for new and improved therapies for HCC. SEC61G is

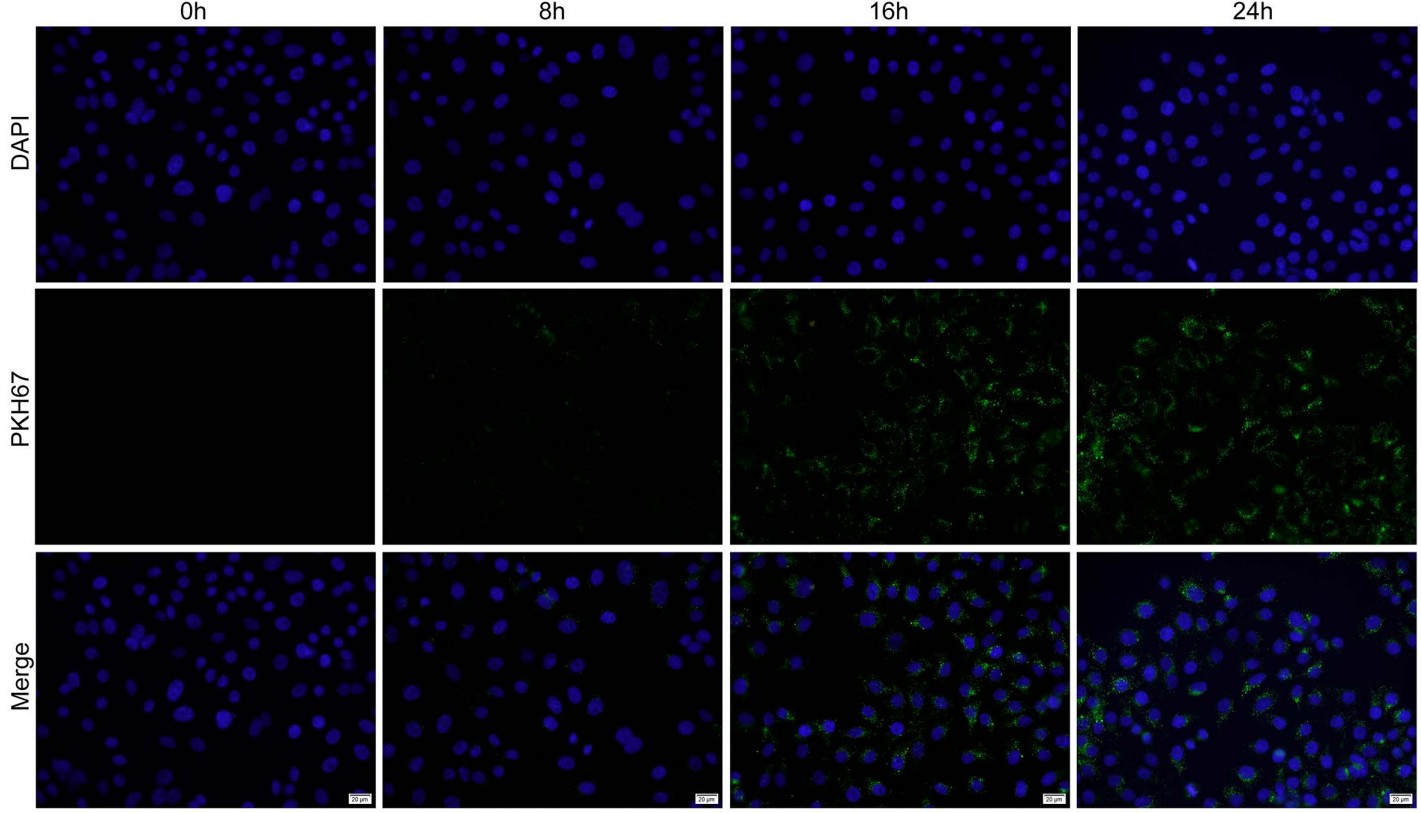

**Fig 3. Exosomes labeling and tracking.** Representative immunofluorescence images of PKH-67-labelled exosomes after co-incubation with HepG2 cells (scale bars: 20 μm).

a subunit of the protein transport protein SEC61, related to the progression, metastasis and prognosis of several tumors [20,25–27]. Moreover, we previously demonstrated that SEC61G was highly expressed in HCC tissue, and its knockdown inhibits HCC tumor cells transformed into a malignant phenotype [24], suggesting that SEC61G is a possible therapeutic target in HCC. Therefore, this study was designed to explore an available way to target SEC61G in HCC, and we demonstrated that miR-488-3p could be transferred by exosomes to target SEC61G that repressed proliferation, migration and invasion of HepG2 cells.

As genetic regulators, miRNAs are uprising therapeutic tools that regulate cellular processes, including HCC progression and metastasis [28–30]. For example, miR-342-3p acted as a tumor suppressor that attenuate tumor development by targeting MCT1, thereby exhibiting promising therapeutic potential in the treatment of HCC [28]. In this study, five miRNAs binding to SEC61G were selected using Targetscan, among which, miR-651-3p and miR-488-3p inhibited HepG2 cells viability and negatively correlated with SEC61G expression. Further dual-luciferase reporter assay confirmed that SEC61G as a target of miR-488-3p in HCC. Therefore, miR-488-3p was selected as a tumor suppressor to demonstrate the therapeutic potential in HCC.

Actually, miR-488 have been demonstrated to play tumor suppressive roles in gastric, pancreatic, breast and skin cancers [31–35]. Expression of miR-488 was reduced in gastric cancer and showed negative correlations with the TNM stage, while miR-488 elevation inhibited malignant cytological behaviors of gastric cancer cells *in vitro* and tumor formation and liver metastasis *in vivo* [31,33]. Down-regulated expression of miR-488 was observed in pancreatic cancer and predicted poor prognosis, while overexpression contributed to apoptosis and G2/M-phase arrest as well

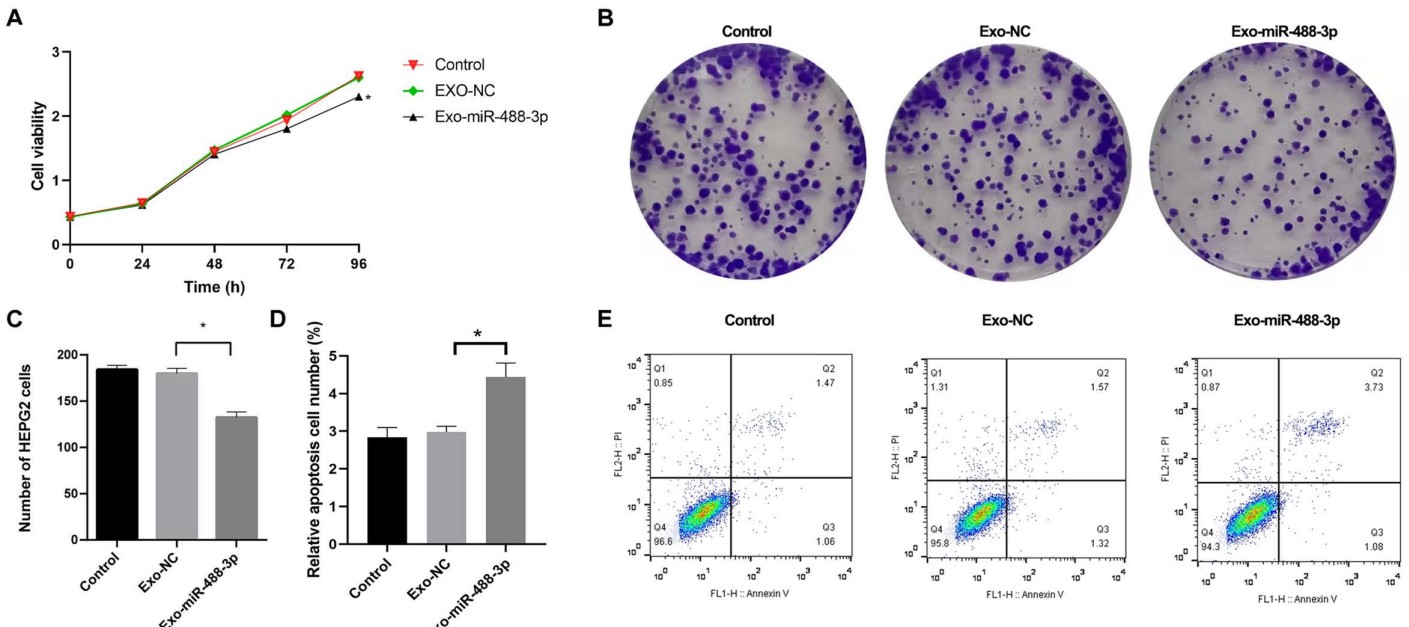

**Fig 4. Effects of Exo-miR-488-3p on proliferation and apoptosis of HepG2 cells.** A: Cell viability of HepG2 cells determined by CCK-8 assay after co-incubated with Exo-miR-488-3p or Exo-NC or controls; Data are presented as the mean±SD from three independent experiments. B and C: Representative images and quantitative analysis of HepG2 cells colony formation after co-incubated with Exo-miR-488-3p or Exo-NC or controls; D: Quantitative analysis of apoptosis. Data are presented as the mean±SD from three independent experiments. *P<0.05. E: Representative flow cytometry plots of apoptotic HepG2 cells.

as inhibited proliferation of pancreatic cancer cells [32]. MiR-488 acted as a tumor suppressor in breast cancer by targeting FSCN1 [35]. Reduced expression of miR-488 was found in metastatic/recurrent colorectal cancer, and its overexpression improved the sensitivity of tumor cells to oxaliplatin/5-Fu [36]. These studies confirmed the therapeutic potential of miR-488 in tumor treatment. However, the roles of miR-488 in HCC have not been fully investigated. Previously, miR-488-5p is known to alleviate hepatic fibrosis [37], and was sponged by exosomal circTMEM181 to regulate CD39 expression in HCC [38]. In this study, miR-488-3p mimic repressed the viability and motility of HepG2 cells, thereby highlighting its therapeutic potential in HCC.

Since miRNA is easily degraded by direct infusion, effective vector-mediated delivery of miR-488-3p may be a novel strategy. Exosomes are a class of membrane-bound nano-sized vesicles that act as intercellular signal modulators to transfer a wide range of biological signals amongst cells [39]. This property as well as their *in vivo* stability and biodegradability allow exosomes to act as carriers for novel treatments [40,41]. Moreover, as drug delivery systems, exosomes have been demonstrated to exhibit unique characteristics, including high biocompatibility, low immunogenicity and toxicity as well as the ability to pass through blood-brain barrier [42]. For example, miR-199a-modified adipose tissue-derived mesenchymal stem cell (MSC) exosomes could effectively deliver miR-199a to tumor cells and improve the sensitivity of tumor cells to doxorubicin in HCC [43]. Exosomes derived from miR-27a-3p transfected MSC could repress proliferation and invasion of HCC tumor cells *in vitro* and tumor metastasis *in vivo* [44]. MiR-26a-modified tumor cell-derived exosomes effectively inhibited the progression of HCC [45]. Similarly, miR-488-3p-loaded engineered exosomes showed anti-tumor effect by significantly hampering viability, colony formation, migration and invasion of HepG2 cells in this study. Therefore, genetically modified exosomes are a possible therapeutic strategy in HCC.

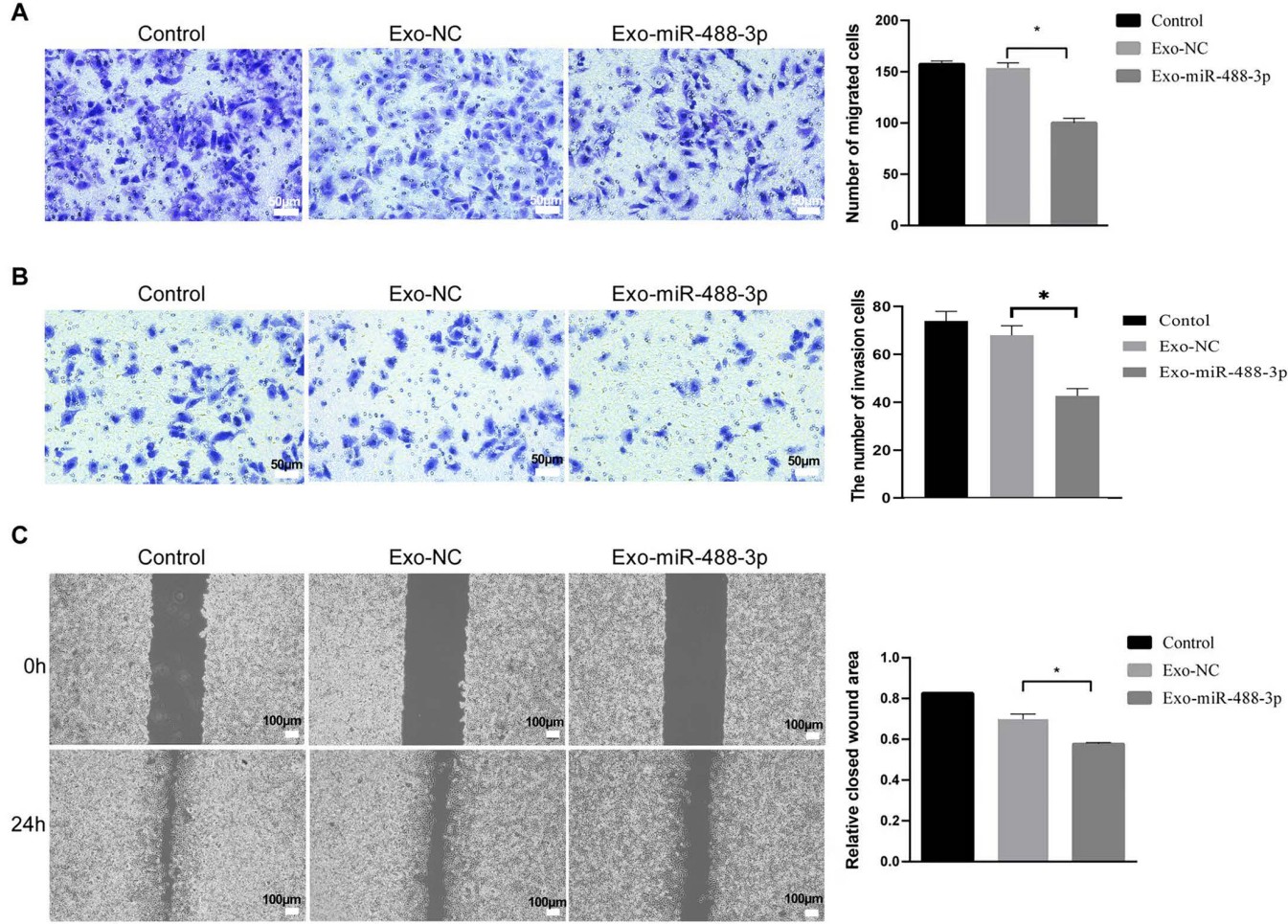

**Fig 5. Effects of Exo-miR-488-3p on migration and invasion of HepG2 cells.** HepG2 cells migration (A) and invasion (B) were determined by Transwell assay after co-incubated with Exo-miR-488-3p or Exo-NC or controls. Data are presented as the mean ± SD from three independent experiments. C, HepG2 cells migration determined by wound-healing assay after co-incubation with Exo-miR-488-3p or Exo-NC or controls. *P < 0.05. Data are presented as the mean ± SD from three independent experiments.

## Conclusion

In summary, miR-488-3p directly targeted SEC61G in HCC, and miR-488-3p-loaded engineered exosomes showed anti-tumor effect by significantly inhibiting growth, migration and invasion of HCC cells. miR-488-3p-loaded engineered exosomes could be an available strategy in the treatment of HCC.

## Supporting information

**S1 Fig. Raw western blot images for SEC61G expression after transfected with miRNAs mimics.**
(TIF)

**S2 Fig. Raw western blot images for exosome markers of CD9, CD81 and HSP70.**
(TIF)

**S1 Table. Raw data.**
(XLS)

## Author contributions

**Conceptualization:** Huijie Gao, Qisen Yan, Zhaoyang Zhang, Cheng Peng.

**Data curation:** Huijie Gao, Zhaobin He, Xiqiang Wang, Qisen Yan, Zhaoyang Zhang, Jun Niu.

**Formal analysis:** Shengbiao Yang, Xiqiang Wang, Naiqing Liu, Zhaoyang Zhang, Jun Niu.

**Funding acquisition:** Cheng Peng.

**Investigation:** Huijie Gao, Chao Gao, Naiqing Liu, Cheng Peng.

**Methodology:** Huijie Gao, Zhaobin He, Shengbiao Yang, Xiqiang Wang, Qisen Yan, Chao Gao.

**Software:** Zhaoyang Zhang, Weibo Niu.

**Supervision:** Weibo Niu, Cheng Peng.

**Validation:** Qisen Yan, Naiqing Liu, Zhaoyang Zhang, Weibo Niu.

**Visualization:** Naiqing Liu, Zhaoyang Zhang, Jun Niu.

**Writing – original draft:** Huijie Gao, Cheng Peng.

**Writing – review & editing:** Jun Niu, Cheng Peng.

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
