## [Decision Letter · Decision Letter 0]

24 Sep 2025

Dear Dr. Peng,

Thank you for submitting your manuscript to PLOS ONE. After careful consideration, we feel that it has merit but does not fully meet PLOS ONE’s publication criteria as it currently stands. Therefore, we invite you to submit a revised version of the manuscript that addresses the points raised during the review process.

We look forward to receiving your revised manuscript.

Kind regards,

Anisha DSouza, Ph.D (Tech)

Academic Editor

PLOS ONE

Journal Requirements:

[This research was funded by the National Natural Science Foundation of China, grant number 82072674; and the Natural Science Foundation of Shandong Province, grant number ZR2020MH258.].

[This research was funded by the National Natural Science Foundation of China, grant number 82072674; and the Natural Science Foundation of Shandong Province, grant number ZR2020MH258.]

[This research was funded by the National Natural Science Foundation of China, grant number 82072674; and the Natural Science Foundation of Shandong Province, grant number ZR2020MH258.]

Additional Editor Comments:

Thank you for your patience and for considering our journal for your interesting manuscript. We have now received the reviewers’ comments, which are included below. Based on their feedback, we kindly request that you revise the manuscript by addressing these points carefully. Please do not hesitate to reach out if you have any questions or need clarification.

Reviewers' comments:

Reviewer's Responses to Questions

**Comments to the Author**

1. Is the manuscript technically sound, and do the data support the conclusions?

Reviewer #1: Yes

Reviewer #2: Yes

2. Has the statistical analysis been performed appropriately and rigorously?

Reviewer #1: Yes

Reviewer #2: Yes

3. Have the authors made all data underlying the findings in their manuscript fully available?

Reviewer #1: Yes

Reviewer #2: Yes

4. Is the manuscript presented in an intelligible fashion and written in standard English?

Reviewer #1: Yes

Reviewer #2: Yes

Reviewer #1: Line 117

The medium name is written as "DEME". Should this be "DMEM"?

Line 128

It might be helpful to include a description of the quantification method used for the isolated exosomes.

Line 171

The product name appears to be outdated. Should this be updated to "ECL Prime"? If the name differs depending on the sales region, forget about it.

Line 247

Just to confirm: Is my understanding correct that miR-488-3p was only effective when loaded into exosomes?

Line 291

For consistency, would it be possible to label the images in Figure 4B and 4E with "Control", "Exo-NC", and "Exo-miR-488-3p", similar to Figure 5?

Reviewer #2: 1. Line 107 and 109- Please can the authors capitalize the whole primer sequence and maintain consistency throughout the manuscript.

2. Line 80- Mention the full form of DMEM and then abbreviate in the manuscript

3. Line 117- DEME is mentioned instead of DMEM

4. Figure 4- Please can the authors label the diagrams B and E with study groups

5. Please can authors provide an explanation why miR-448-3p activity and effectiveness was not looked at throughout the study?

6. Please can the authors mention how many times the experiments were performed in the figure legend or results

7. Figure 1- Is it possible for the western blot to be repeated, the miR-375-3p has two bands, please can you provide an explanation.

8. Figure 4 A- cell viability just has two lines for 3 groups, can you change it to color or make the third group prominent.

9. Figure 4C- there is no conclusive evidence that the apoptosis numbers were elevated since there is no statistical significance, the authors can confirm this by performing western blot for apoptotic markers like Bax, BCl2, PARP

10. Figure 5- scale bars are missing in the image. Also, some graphs are bordered and some are not, please try to maintain consistency throughout the figures.

**Do you want your identity to be public for this peer review?** For information about this choice, including consent withdrawal, please see our Privacy Policy

Reviewer #1: No

Reviewer #2: No

---

## [Author Response · Author response to Decision Letter 1]

2 Dec 2025

Dear editor and reviewer

Thanks for your kind letter about our manuscript (PONE-D-25-40839) entitled “MicroRNA-488-3p-loaded engineered exosomes inhibit proliferation, migration and invasion of hepatocellular carcinoma by targeting SEC61G”. We wish to extend our sincere gratitude to you for the time and effort spent on reviewing our manuscript and for providing valuable feedback. We are greatly encouraged by the positive and constructive comments.

We are delighted that you found the manuscript to be technically sound, statistically rigorous, well-presented.

We have carefully considered all the comments and suggestions . In the point-by-point response below, we have addressed each comment in detail and have incorporated all suggested changes into the revised manuscript. The revisions are highlighted in the revised manuscript using the “Track Changes” feature.

We believe that addressing these comments has significantly strengthened the manuscript, and we hope that the revised version now meets the approval of the editorial board and the reviewers.

Sincerely,

Cheng Peng

---

## [Decision Letter · Decision Letter 1]

22 Dec 2025

Dear Dr. Peng,

Thank you for submitting your manuscript to PLOS ONE. After careful consideration, we feel that it has merit but does not fully meet PLOS ONE’s publication criteria as it currently stands. Therefore, we invite you to submit a revised version of the manuscript that addresses the points raised during the review process.

We look forward to receiving your revised manuscript.

Kind regards,

Anisha DSouza, Ph.D (Tech)

Academic Editor

PLOS One

Journal Requirements:

Additional Editor Comments:

Thank you for addressing the comments. One reviewer has provided a very minor typographical suggestion. Please address it and share the revised manuscript for the next steps. Thank you.

Reviewers' comments:

Reviewer's Responses to Questions

**Comments to the Author**

Reviewer #1: (No Response)

Reviewer #2: All comments have been addressed

2. Is the manuscript technically sound, and do the data support the conclusions?

Reviewer #1: Yes

Reviewer #2: Yes

3. Has the statistical analysis been performed appropriately and rigorously?

Reviewer #1: Yes

Reviewer #2: Yes

4. Have the authors made all data underlying the findings in their manuscript fully available?

Reviewer #1: Yes

Reviewer #2: Yes

5. Is the manuscript presented in an intelligible fashion and written in standard English?

Reviewer #1: Yes

Reviewer #2: Yes

Reviewer #1: line 175

Please ensure that the trademark symbol in “ECL-plus™ kit” is formatted as a superscript (™) according to standard typographic conventions. This is a minor formatting issue but should be corrected for consistency.

Reviewer #2: (No Response)

**Do you want your identity to be public for this peer review?** For information about this choice, including consent withdrawal, please see our Privacy Policy

Reviewer #1: No

Reviewer #2: **Yes:** Advait Shetty

---

## [Author Response · Author response to Decision Letter 2]

30 Dec 2025

Dear Reviewers,

Thank you for reviewing our manuscript (PONE-D-25-40839R1) entitled “MicroRNA-488-3p-loaded engineered exosomes inhibit proliferation, migration and invasion of hepatocellular carcinoma by targeting SEC61G” and for your positive feedback. We are pleased that you found our previous revisions satisfactory. We have carefully addressed the comment below and have prepared the revised manuscript accordingly.

We believe that the manuscript now fully complies with PLOS ONE’s formatting and editorial standards. Thank you once again for your valuable time and guidance throughout the review process.

Sincerely,

Cheng Peng

---

## [Editor Report · Decision Letter 2]

2 Jan 2026

MicroRNA-488-3p-loaded engineered exosomes inhibit proliferation, migration and invasion of hepatocellular carcinoma by targeting SEC61G

PONE-D-25-40839R2

Dear Dr. Peng,

We’re pleased to inform you that your manuscript has been judged scientifically suitable for publication and will be formally accepted for publication once it meets all outstanding technical requirements.

Kind regards,

Anisha DSouza, Ph.D (Tech)

Academic Editor

PLOS One

Additional Editor Comments (optional):

Thank you for addressing the typographical errors.
---

## [Editor Report · Acceptance letter]

PONE-D-25-40839R2

PLOS One

Dear Dr. Peng,

I'm pleased to inform you that your manuscript has been deemed suitable for publication in PLOS One. Congratulations! Your manuscript is now being handed over to our production team.

Kind regards,

on behalf of

Dr. Anisha DSouza

Academic Editor

PLOS One